# Objective and Perceived Neighborhood Greenness of Students Differ in Their Agreement in Home and Study Environments

**DOI:** 10.3390/ijerph17103427

**Published:** 2020-05-14

**Authors:** Alexander Karl Ferdinand Loder, Josef Gspurning, Christoph Paier, Mireille Nicoline Maria van Poppel

**Affiliations:** 1Institute of Sport Science, University of Graz & Staff Department Quality Management, University of Music and Performing Arts Graz, 8010 Graz, Austria; 2Institute of Geography and Regional Sciences, University of Graz, 8010 Graz, Austria; josef.gspurning@uni-graz.at; 3Institute of Sport Science, University of Graz, 8010 Graz, Austria; christoph.paier@edu.uni-graz.at (C.P.); mireille.van-poppel@uni-graz.at (M.N.M.v.P.)

**Keywords:** neighborhood greenness, environmental psychology, public health, sedentariness, green space, built environment, natural environment, Austria

## Abstract

Research has reported the associations between objective or subjective neighborhood greenness and health, with low agreement between the greenness scores. College students are prone to poor health, and data are lacking on home and university environments. We studied the agreement between greenness parameters and the associations of objective greenness with health in different locations. Three hundred and seventy-seven college students were recruited, with a mean age of 24 years, in the city of Graz, Austria. Objective and perceived greenness was assessed at home and at university. Health measures included the WHO-5 questionnaire for mental health, the IPAQ questionnaire (short) for physical activity and sedentariness, and body mass index. Per location, quintile pairs of objective and perceived greenness were classified into underestimates, correct estimates or overestimates. Interrater reliability and correlation analyses revealed agreement between greenness scores at home but not at university. ANOVA models only showed poorer mental health for students underestimating greenness at university (*M* = 51.38, *SD* = 2.84) compared to those with correct estimates (*M* = 61.03, *SD* = 1.85). Agreement between greenness scores at home but not at university was obtained, and mental health was related to the perception of greenness at university. We conclude that reliable and corresponding methods for greenness scores need to be developed.

## 1. Introduction

The biophilia hypothesis states that humans have developed and maintained an affinity for nature throughout their evolution [1,2]. Although there are plans and best practices for reintroducing nature into areas with a high degree of built environment [3], ongoing urban densification in cities is linked to a decline in natural features and “green space” [4,5]. Green space is operationalized differently in distinct research contexts, but most definitions have in common that it includes vegetation, such as trees, bushes, grass and areas of water [6]. Despite the infrastructure for a technologically advanced world, humans need a certain “dose” of nature in their lives for their wellbeing, according to the biophilia hypothesis [1]. A growing body of research shows that neighborhood greenness in cities is an integral component of residents’ health; there are positive associations between green space and mental health [7] and physical activity [8,9,10,11,12] as well as negative associations with sedentariness [13] and body mass index (BMI) [14,15].

The assumption, the mere existence of nature around people’s residences influences health, does not incorporate human information processing. A second dimension therefore needs to be addressed, namely, how the real (objective) environment is subjectively perceived. The idea that the degree of perceived nature in a resident’s immediate neighborhood is equally important for certain health parameters arises from studies incorporating both objective and subjective measures of greenness in their designs [16,17,18,19]. Furthermore, perceived greenness may act as a mediator between objective greenness and health due to different outcomes depending on objective or subjective measurements [17]. However, one study found positive associations for subjective measures of greenness and the number of walking trips in a given timeframe, which could not be observed for objective measures [16]. This implies that there can be a discrepancy between the objective greenness measures and the subjectively perceived nature in the environment [19,20]. Evidence in this direction comes from a study indicating that overweight or less physically active people (for transportation purposes) with lower education and lower incomes are more likely to perceive a high walkable neighborhood as low walkable [21].

### 1.1. Neighborhood Greenness in Different Environments

Most previous studies have assessed the effects of greenness in the neighborhood of residents’ homes or unspecified locations [7,22], although people do not only spend time at home over the course of a day. Many adults spend several hours of their average day at work, and many students attend university or spend time at other educational facilities during the day. Less research has focused on the effects of the environment depending on locations such as the workplace, university campuses, schools, hospitals or retirement homes [23,24,25,26,27]. However, connections between the perceived greenness levels of a university campus and the self-reported quality of life of college students have been found [24]. Another study found associations of objective and perceived greenness at different locations on a campus with perceived restorativeness and quality of life [28]. These results hint at the importance of investigating the effects of the greenness of the environment in different locations. Compared to the neighborhood at home, the campus environment is not likely to be chosen by students due to its natural features. This may lead to different outcomes for mental health, physical activity, sedentariness and BMI, as well as to a different agreement between objective and perceived greenness dependent on the location.

### 1.2. Objective

This study aims to gain new insights into the agreement between the objective and perceived neighborhood greenness for college students at home and at university. As a second aim, we want to explore the association of objective neighborhood greenness in both locations with mental health, physical activity, sedentariness and BMI in students.

## 2. Materials and Methods

This study is part of a larger research project, which collected data via an online survey between October and December 2017. For the current paper, we investigated the agreement between objective and subjective greenness measures as well as their importance for different health parameters.

All study participants had their homes and study environments in and around the city of Graz (the capital of Styria, Austria). Graz is predominantly known for its university area, has approximately 300,000 residents and spans 128 km^2^. The neighboring campuses of the three major universities (University of Graz, the University of Music and Performing Arts, and the Technical University) form this university district. There is only little distance between the respective campus areas, which is why quite some students travel frequently between the facilities due to the range of interdisciplinary and interuniversity study programs. In Graz, students do not live on the campus area; they have their homes elsewhere in the city or surrounding areas. Due to these characteristics, Graz was considered a good place for representative student data in (mid-)Europe. This research project has been approved by the ethics committee of the University of Graz (GZ 39/58/63 ex 2016/17).

### 2.1. Study Population and Data Collection

How data was acquired has been described in two previous papers [13,29]. In short, participants were recruited via social media and university-related mail distribution lists. Therefore, the total number of people that were invited to participate is unknown. From 758 respondents, who took part in the study, 115 were removed due to missing informed consent, another 248 due to a lack of objective greenness data and another 18 due to having an employment status different from “college student”. The sample analyzed in this study consisted of 377 students.

### 2.2. Study Variables and Questionnaires

#### 2.2.1. Objective Greenness Measures

From a spatial science perspective, the attempt to quantify greenness measures—with as little subjective bias as possible—is based on the recording of land-cover/land-use features’ spatial distributions, which are interpreted as relevant for greenness in the context of this study (e.g., bushes, meadows, forests, individual trees, public parks and gardens) [6]. Special attention was paid to the fact that these entities, which are decisive for the result, are actually “optically effective”, i.e., can be seen by the participants. For this reason, flower gardens behind high walls are irrelevant for the study, which is why—first and foremost—a methodical concept had to be developed, which, with the help of a geographic information system (GIS), should enable the calculation of real visible greenness units. For better comparability, these calculated values are presented as area shares of the total area of the examined area.

In the online survey, participants were asked to allocate and mark their approximate home and study environments on an interactive map. They were instructed to place the map pin somewhere near each location (at least in front of the relevant building). They were not asked to indicate their exact address, since this might have decreased compliance due to data security concerns. In order to derive a reasonable buffer zone within which they were expected to be active around each location, they were asked how often they used different modes of transport. As was to be expected given the composition of the sample, the participants primarily used bicycles and public transport. Based on this finding, a zone of activity-buffer was chosen to define the area used for a realistic determination of the objective green value. This allowed both bicycles and public transport to be considered as means of transport. The research project did not collect data about the daily bicycle routes of the participants, and since the possible entry points into the road network system in Graz are ubiquitously distributed, the parameterization of the buffer width was primarily oriented towards the public transport stops. Based on the maximum values for both the nearest stops for each individual’s home (M = 154.00 m, SD = 69.22 m, Max = 299.90 m) and university (M = 128.17 m, SD = 56.65 m, Max = 296.35 m), a buffer zone of 300 m for both locations was defined. This range is in line with the recommendation of the European Commission that open public spaces should be within a distance of 300 m around residences [30]. Additionally, it matches the buffer-widths of several other studies, as listed in a literature review [31]. Single extreme values were interpreted as irrelevant outliers and therefore removed beforehand.

For the implementation of the spatial analysis concept in a GIS, geo data of different origins—official as well as semi-official—had to be used to enable all the analyses required. This procedure necessitated a data homogenization step preceding the core analyses to ensure the quality of the results. The base map data were accessed via OpenStreetMap Sources over the internet [32]. In order to derive as many reliable land-cover/land-use data as possible, the administration of the city of Graz was contacted to get the data for delineating the vegetation indices of the city area. We received shapefiles for processing in geospatial software, already limited to the predefined 300 m buffer-zone around each data point. Due to data security concerns, we did not receive data beyond this range. The attribute table of this database covers the attribute values, built space, green space, bushes, trees, water (blue space) and other built characteristics as a summary value (e.g., sidewalks). After removing built environment and other built entities—as already mentioned—the percentage was summarized for the area covered by all the natural features within the buffer zones, which was then used as our measure of objective greenness. Since not every participant was living or studying within the area that was covered by the dataset from the administration of the city of Graz, incomplete cases were removed for the analyses. This resulted in 303 complete cases that were left for objective greenness at home and 367 complete cases for university (377 total).

#### 2.2.2. Perceived Greenness Measures

Perceived greenness measures were previously described in detail [13]. Measures were obtained from six questions on the accessibility, presence and quality of greenness in both the home and study environments, on a scale of 0 to 5 (very good; good; neither good nor bad; bad; very bad) [33,34]. Each of the following questions had to be answered two times, for the home as well as the campus environment (twelve questions in total):

How would you describe:Access to parks or nature?Access to walking or bicycle paths?The presence of greenness?The presence of trees/tree density or canopies along footpaths?The presence of other natural features?Overall, how would you describe the quality of the green/blue space at your location?

Means of all the answers per location were calculated and then transformed into percentages, which led to a measure for perceived greenness at home and at university. In order to make a comparison with objective greenness, a variable “perceived presence of greenness” was created from the means of the three items focusing on the presence of greenness, trees/tree density or canopies and other natural features per location.

#### 2.2.3. Comparison of Objective and Perceived Greenness Scores

The objective and perceived greenness measures differed from each other in their characteristics, i.e., the responses to questions about the subjective accessibility, quality and presence of greenness vs. the objective presence of greenness. Although the perceived presence of greenness was added as a parameter more similar to objective greenness, there are still differences in how these variables were measured. Therefore, objective and perceived greenness measures were divided into quintiles, resulting in five almost equally sized groups for each variable (from the first quintile with low greenness to the fifth quintile with high greenness), as shown in Appendix A. Subsequently, the quintiles for objective greenness and perceived greenness as well as for objective greenness and the perceived presence of greenness were subtracted from each other to see whether scores differed by more than one quintile. This resulted in four variables, for the home and study environments, with people who underestimated (more than one quintile for perceived greenness under the quintile for the objective measure), who correctly estimated (variation within one quintile upwards or downwards) and who overestimated greenness (more than one quintile for perceived greenness over the quintile for the corresponding objective measure).

#### 2.2.4. Mental Health

Mental health was assessed with the German version of the WHO (Five) Well-Being Index, which focuses on the last two weeks of wellbeing [35,36].

#### 2.2.5. Physical Activity

Physical activity was assessed with the German version of the short-form of the IPAQ questionnaire [37]. The activity measures of the questionnaire were transformed into MET-min per week, summed up into a single measure for physical activity and transformed into MET-h per week [38]. Outliers (values > 400 MET-h per week) were removed. Sedentariness was the reported number of hours spent sitting on a usual weekday.

#### 2.2.6. Body Mass Index

Participants were asked about their height and body weight to obtain the BMI score of each individual (calculated as weight in kg over m^2^).

### 2.3. Statistical Analysis

Correlational analyses were conducted via Pearson correlational analyses. They were used to observe to which extent the scores of the objective greenness values at home and at university correlated with the perceived greenness values at home and at university. Tests for collinearity were conducted using Pearson correlations. The interrater reliability of the estimates of the true degree of objective greenness around participants’ homes and university was assessed using Cohen’s kappa. The association of underestimates, correct estimates or overestimates of green space with other participant characteristics (mental health, BMI and physical activity) was analyzed using univariate two-way ANOVA models. Two-way multivariate mixed-ANOVA models were used to adjust the objective and perceived greenness scores for gender and marital status. Post hoc comparisons were accomplished via Bonferroni corrections. Linear regression models were used to assess the associations between objective greenness scores and the relevant outcomes.

## 3. Results

### 3.1. Sample

The study sample consisted of 377 college students, 281 (75%) of whom were female, with a mean age of 24 (*SD* = 7) years. Additional characteristics of the sample can be found in Table 1.

### 3.2. Collinearity

The collinearity checks are summarized in Appendix A and Appendix A. With exception of the correlations between the perceived presence of greenness and perceived greenness scores, the strongest correlation found was between age and income (*r* = 0.54, *p* < 0.001). Tolerance statistics and variance inflation factors indicated that collinearity was not of concern.

### 3.3. Objective and Perceived Neighborhood Greenness

The perceived greenness scores for both the home environment (*M* = 79.49%, *SD* = 16.40%) and the university environment (*M* = 69.98%, *SD* = 18.00%) as well as the scores of the perceived presence of greenness at home (*M* = 76.83%, *SD* = 19.71%) and at university (*M* = 65.83%, *SD* = 20.72%) were high, in general, compared to objective greenness at home (*M* = 27.52%, *SD* = 11.07%) and at university (*M* = 20.78%, *SD* = 7.41%). Pearson correlations yielded a significant but small-to-moderate correlation between objective and perceived greenness at home (*r* = 0.23, *p* < 0.001). However, no correlation between objective and perceived greenness at university was found (*r* = −0.05, *p* = 0.333). The perceived presence of greenness at home correlated significantly with objective greenness at home (*r* = 0.25, *p* < 0.001), but there was no correlation between those measures at university (*r* = −0.05, *p =* 0.366). Significant correlations were obtained for the perceived greenness at home and perceived presence of greenness at home (*r* = 0.96, *p <* 0.001) as well as for the perceived greenness at university and perceived presence of greenness at university (*r* = 0.96, *p <* 0.001).

Similarly, there was significant agreement between the estimates of objective greenness at home (based on quintiles) and perceived greenness at home (*κ* = 0.11, *p* < 0.001). No agreement was found for the estimates of objective greenness at university and perceived greenness at university (*κ* = 0.01, *p* = 0.744). Figure 1 shows the means and standard deviations of the perceived greenness measures per quintile of objective greenness. Agreement was slightly higher for presence of greenness at home (*κ* = 0.15, *p* < 0.001), but there was also no agreement for the presence of greenness at work (*κ* = −0.05, *p* = 0.052).

### 3.4. Estimation of Greenness and Health

Mental health differed significantly between people who estimated greenness in their university environment more or less accurately. Post hoc comparisons with Bonferroni corrections showed that participants that underestimated greenness at university had poorer mental health (*M* = 51.38, *SD* = 2.84) compared to those that correctly estimated greenness at university (*M* = 61.03, *SD* = 1.85, *F*(2, 284) = 4.06, *p* = 0.014). No differences were observed between people who correctly estimated or overestimated greenness at university, nor between those who underestimated or overestimated. No differences were found between groups for the estimation of greenness at home. The results of the ANOVA models can be found in the Supplementary Material (Appendix A).

Two-way ANOVAs did not show differences in physical activity (MET-h), sedentariness (hours spent sitting per day) or BMI between people who underestimated, correctly estimated, or overestimated greenness at home and at university. The results did not change when the ANOVA models included gender, age, income, education and marital status as covariates (data not shown).

Two-way multivariate mixed ANOVA models with objective and perceived greenness as dependent measures showed no significant differences according to gender or marital status (Appendix A).

### 3.5. Associations with Objective Greenness

No significant associations between objective greenness and mental health, physical activity, sedentariness or BMI were found. The results can be found in Table 2. These associations did not change when adding the potential confounders of gender, age, income, education and marital status to the models (data not shown).

## 4. Discussion

### 4.1. Main Findings

In this study, we aimed to analyze the agreement between objective and perceived greenness measures at home and at university and to compare the mental health, BMI and physical activity of people who underestimated, correctly estimated or overestimated greenness in those distinct places.

In line with our expectations, we found a small-to-moderate correlation and agreement between objective and perceived greenness at home. However, no correlation or agreement was found between objective and perceived greenness at university. This last finding corresponds with previous research reporting a lack of agreement between objective and subjective greenness scores [19,20]. A possible explanation for the difference in agreement between home and university could be that people are more familiar with their home environment, leading to a better estimation of the greenness at home.

People who underestimated, correctly estimated, or overestimated the greenness at home or at university were not different with regards to their physical activity level or BMI. However, people with poorer mental health were more likely to underestimate greenness at university but not at home.

### 4.2. Objective and Perceived Greenness at Home and at University

Most of the prevailing literature on green space does incorporate greenness in distinct locations [7,22]. This study demonstrates that location matters. We only found some agreement between greenness measures at home but not at university. Since opportunities for direct and prolonged exposure to nature may be higher at home than at university, it may be easier for people to estimate the true extent of natural features in their home neighborhood than around the university. A higher number of repeated and more extensive trips into the environment at home may lead to an enhanced spatial memory of this area compared to the study environment [39]. Knowledge of what the outdoor environment of the campus actually looks like might be limited beyond the daily movement radius for reaching the facility and its buildings and beyond the direct view from one’s lecture hall. Research points out the importance of being able to visually experience environments for spatial cognition [40]. In line with this, residential tenure might be an important measure to control for in future studies.

Our results might have been influenced by the difference in the greenness measures taken, since perceived greenness scores included the accessibility, quality and presence of greenness, and the objective measurements could only reflect the presence of greenness. However, analyses with the variable “perceived presence of greenness” did not change the results. High correlations between the perceived greenness and perceived presence of greenness show that a large part of the variance is shared between those variables.

The authors of another study that focused on both objective and subjective greenness argued that the lack of agreement between those values is to do with how measures are assessed [19]. Objective greenness is translated into numerical data by the area that is covered by natural features on a 2D map surface (as seen from above) within predefined buffer zones. Not only does this lead to a loss of the third dimension (height), it also neglects what is visible on the ground for study participants. A green garden on top of a high building or on the other side of a high wall results in the same objective greenness values as if it were accessible/visible at ground level. Additionally, a leaf canopy could cover either meadows and bushes or roads and sidewalks, which cannot be distinguished in the objective data. Therefore, it is to be expected that the ratings of perceived greenness differ from objective scores depending on which features can be seen and accessed by residents. In addition, it depends on the questions used for measuring the perceived quality of green space [19]. However, more research is needed regarding differences in perceived greenness depending on other characteristics of a person, such as mental health.

### 4.3. Objective and Perceived Greenness and Mental Health

People who underestimated greenness at university had poorer mental health compared to people who correctly estimated this. Previously, we reported an association of the perceived greenness of both the home and university environments with mental health [28]. However, we did not find an association of mental health with objective greenness, either at home or at university. This might indicate that mental health might affect the perception of the environment, or vice versa. Our finding also adds evidence to the notion that perceptions of the environment may be more or equally predictive of certain variables as the objectively measured environment [16]. We only found significant differences in mental health between people underestimating and correctly estimating greenness at university but no differences for the estimations of greenness at home. There may be an underlying bidirectional association between stress coping and potential stressors in the locations we studied. A higher potential for stress at university compared to in home environments in general and differences in the mental health of people with good and bad coping strategies may explain the results we obtained [41]. An influence of natural “pessimists” and “optimists” skewing the data is not likely. Only a small to moderate correlation was present between perceived greenness at home and that at university. This argues against such an influence, as it can be expected that general good or bad ratings would increase the strength of this correlation. However, these assumptions need further study.

### 4.4. Strengths and Limitations

Highly green environments in cities, which are more favorable residential environments, are rather chosen by people who can afford to live there [42]. In line with this, it can be expected that socioeconomic status and income influence the outcomes of studies not controlling for these dimensions. The student sample of this study is homogenous with respect to age, socioeconomic status, income and education. Due to this makeup, the majority of the participants had a high education level and an income lower than 1000 € per month, which made confounding by socioeconomic dimensions unlikely. Additionally, this makes the sample not representative of all adult residents in and around the city of Graz. Our BMI measures cannot be reliably interpreted in the context of overweight and obesity outside of clinical contexts; they do not account for muscle mass and bone mineral density.

The method used for measuring objective greenness has its shortcomings, which are primarily due to the available data material and the resulting weaknesses. If one accepts the basic concept of quantifying greenness by means of the greenness component in relation to the (recognizable) overall environment, two main problems remain to be solved. First, the question of which characteristics that can be depicted in spatial databases make an evaluation in the sense of the present study possible at all needs to be answered; in other words, a consensus has to be created regarding the attributes of greenness, whereby these have to be collected and entered into the spatial databases. The second problem deals with spatial perception; apart from the generally inadequate consideration of the third dimension (height), as mentioned earlier, aspects essential for the study have not been considered at all so far—the limitation of the area able to be seen/recognized by the human field of vision and the effective range of our vision. Finally, the different cognitive abilities of the individual participants should also be taken into account.

This study did not collect data on indoor environments. However, positive health outcomes can be expected from green indoor environments, such as reduced air pollution, which is partially mediated by the greenness of the outdoor environment [25]. Similarly, plants at the workplace are capable of reducing stress and anxiety [43]. The makeup of indoor environments may thus be an important confounder in people’s homes and study environments.

### 4.5. Prospects

Based on the results of the current study, future research could entail a more detailed look at the association between both objective and perceived greenness in the context of different environments, such as at home, places of study or workplaces. Further research is needed to explore the implications of this approach. In line with this, more valid and reliable methods are needed to uncover the “true” relationships between objective and subjective greenness. This means that study designs including objective measurements need to take into consideration how study participants perceive and describe their environments at eye level [44]. For instance, recent research using publicly available street-view images found associations between the quality and quantity of street greenery and residents’ physical activity levels [45]. Two clear priorities can be identified, one more data-centered and one more method-centered. While the former, however, requires the consensual development of a suitable attribute data set and thus the (usually cost-intensive) new collection of all data relevant to the study, the (GIS) methodological improvement can be achieved relatively easily with the help of suitable infrastructure. Experiments carried out in parallel to the present study, e.g., to delineate the environmental areas actually perceived by cyclists, already show relatively promising results.

In addition, questionnaires and other methods for assessing perceived natural features also have to be refined and adjusted in a direction that is in line with what is currently technically possible for obtaining objective values.

In order to develop effective public health strategies, the drivers of the association between greenness and health should be of major interest. Therefore, it is important to assess whether the subjective perception of greenness, instead of objective greenness, is the most important factor in this association. If so, we need to better understand which features of objective greenness have the greatest impact on peoples’ perception of nature, in order to improve their health outcomes.

## 5. Conclusions

This study found positive associations and agreement in the ratings between objective and perceived greenness for home but not for university environments. Students underestimating greenness levels at university showed poorer mental health scores compared to those who correctly estimated greenness at university. The results indicate that more attention should be paid to developing validated, reliable and corresponding methods for assessing both objective and perceived greenness scores. Future studies are needed in more diverse populations and in other geographic locations.

## Figures and Tables

**Figure 1 ijerph-17-03427-f001:**
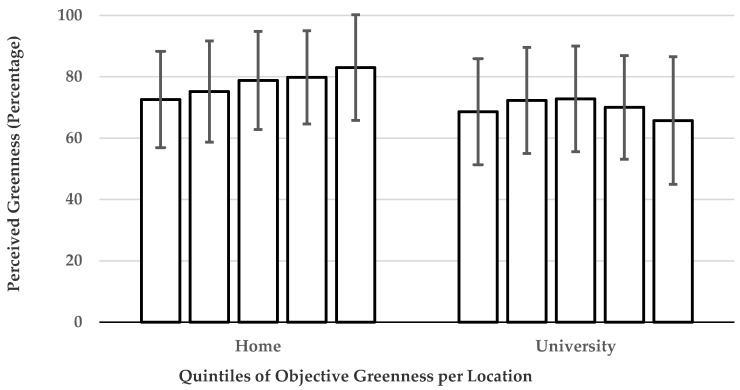
Mean of perceived greenness scores per quintile of objective greenness at home.

**Table 1 ijerph-17-03427-t001:** Sample characteristics.

Variable	Value(s)
	*n* = 377
Gender	281 (75%) female 87 (23%) male 9 (2%) missing
Marital Status	14 (4%) married 157 (42%) in a partnership 181 (48%) single 25 (7%) missing
Income per Month	238 (63%) less than 1000 € 57 (15%) between 1001 € and 3000 € 6 (2%) more than 3001 € 76 (20%) missing
Living in Graz	314 (83%) objective greenness measures: 303 (80% of *n*)
Studying in Graz	367 (98%) objective greenness measures: 367 (97% of *n*)
Age in Years, *M* (*SD*)	24 (7)
Objective Greenness at Home, *M* (*SD*)	27.52% (11.07%)
Objective Greenness at University, *M* (*SD*)	20.78% (7.41%)
Perceived Greenness at Home, *M* (*SD*)	79.49% (16.40%)
Perceived Greenness at University, *M* (*SD*)	69.98% (18.00%)
Perceived Presence of Greenness at Home, *M* (*SD*)	76.83% (19.71%)
Perceived Presence of Greenness at University, *M* (*SD*)	65.83% (20.72%)
Greenness Estimation (Quintiles) for Home	Underestimation: 71 (23%) Estimation within one quintile: 182 (60%) Overestimation: 50 (17%)
Greenness Estimation (Quintiles) for University	Underestimation: 93 (25%) Estimation within one quintile: 190 (52%) Overestimation: 83 (23%)
Greenness Estimation (Quintiles) for the Presence of Greenness at Home	Underestimation: 68 (18%) Estimation within one quintile: 185 (49%) Overestimation: 50 (13%)
Greenness Estimation (Quintiles) for the Presence of Greenness at University	Underestimation: 78 (21%) Estimation within one quintile: 186 (49%) Overestimation: 102 (27%)

**Table 2 ijerph-17-03427-t002:** Associations between objective greenness and relevant outcomes of linear regression models.

Parameter	R^2^	F	df	b	t	df	95% CI b	*p*
Mental Health								
At Home	0.002	0.63	1, 301	−0.08	−0.77	301	−0.27–0.12	0.440
At University	< 0.001	0.001	1, 365	−0.004	−0.03	365	−0.27–0.26	0.973
Physical Activity in MET-h								
At Home	0.002	0.69	1, 297	−0.02	−0.83	277	−0.06–0.02	0.408
At University	0.002	0.89	1, 360	0.03	0.94	360	−0.03–0.08	0.346
Sedentariness in hours spent sitting per day								
At Home	0.01	2.03	1, 299	0.02	1.42	299	−0.01–0.06	0.156
At University	0.001	0.20	1, 360	−0.01	−0.44	360	−0.06–0.04	0.659
BMI								
At Home	0.002	0.67	1, 297	−0.02	−0.83	297	−0.06–0.02	0.408
At University	0.002	0.89	1, 360	0.03	0.94	360	−0.03–0.08	0.346

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
