# Peer review of "Objective and Perceived Neighborhood Greenness of Students Differ in Their Agreement in Home and Study Environments"

_ijerph, 2020, doi:10.3390/ijerph17103427_

Round 1

Reviewer 1 Report

This study examined associations between “objective or subjective” greenness and health. The manuscript possesses the potential to inform urban design standards that could theoretically advance population health and wellbeing, and the authors are commended for taking a novel approach to examining the complex relationship between objective and subjective greenness and mental and physical health. However, there appear to several limitations to the manuscript.

  1. The introduction would benefit from clarification of key terms.
    1. For example, on line 35, the authors introduced the concept of “green space,” yet neglect to identify what constitutes greenspace until the methods section. This should be clarified upfront and grounded in the available literature to avoid any confusion between arguments made in the introduction and the selected research approach. Furthermore, the authors operationalized greenspace in the methods section as “bushes, meadows, forests, individual trees,” but seem to miss public parks, gardens, etc.
    2. In addition, the authors contend that “urban densification is responsible” for greenspace and natural feature declines (lines 34-35), yet many cities with high population densities have moved toward green rooftops, greenways, community gardens, etc. (see New York City, for example) to reintroduce greenspace, thus increasing greenspace. A more thorough review of the literature would be helpful.
    3. On line 42, the authors contend that connections between nature (are they are referring to greenspace?) and health are assumed. However, the prior paragraph cites a “growing” body of literature. In fact, there is a wealth of literature on this topic. Therefore, framing this as an assumption seems peculiar.
  2. The literature review does not provide sufficient evidence for key arguments.
    1. The Perceived Neighborhood Greenness section should review a broader body of literature.
      1. For example, the authors indicate that objective and subjective measure should be utilized when examining health benefits, yet only cite one project.
      2. On lined 49-50, the authors state that “studies” indicate disagreement between objective and perceived greenness and walking, but only cite one paper. Please include additional evidence if referring to multiple studies.
    2. The Neighborhood Greenness in Different Environments section would also benefit from a more extensive literature review.
      1. The authors indicate that “most” research concerns home neighborhoods and unspecified areas. They seem to neglect the body of literature that examines connections to greenspace in other contexts. I would be helpful to review this in addition the two papers they cite concerning campus environments.
    3. Finally, there are several awkward sentences in this section that would benefit from additional editing.
  3. The methods section would benefit from additional clarification.
    1. Please note there are several awkward sentences that make this section difficult to read.
    2. Please operationalize “greenness” upfront.
    3. It may be helpful to include a table of all “greenness” features included in the study.
    4. As the authors indicated that public transportation and bicycles were the primary means of transportation, the rationale for a 300m buffer of public transportation stops needs to be further explicated. Citing the literature would be helpful.
    5. Other researchers may wish to replicate this study, so it would be helpful to further explain methods for delineating greenspace within 300m buffers. For example, were these shapefiles that could be examined in geospatial software?
    6. While perceived greenness measures are published elsewhere, since they are critical to this study, it would be helpful to include here.
    7. On line 138, the authors indicate that their survey concerned perceived “quality and accessibility.” However, the more pertinent language on the survey concerns “presence,” as the objective measure seems to measure presence of absence.
    8. A major methodological flaw is that that the authors appear to measure very different constructs with the objective (i.e., presence or absence of “greenness”) and perceived (quality, accessibility, presence of “greenness”) measures of greenness. This approach requires further clarification, as the objective measures do not account for quality and accessibility, only whether there were green features within 300m of stated locations. The authors do recognize this on lines 142-143, but the method for rectifying this (quintiles) does not address the major issue. I might suggest that they only use the presence score from the “perceived” measures. At the very least, provide some evidence (literature) for using the stated methodology.
    9. Please not that calculating BMI based on height and weight does not account for muscle mass and bone density, and thus should not be used as an indicator of overweight/obesity outside of the clinical context.
    10. Did the authors investigate participant tenure at their current residence? This would be an important confounder.
    11. Please explain why linear regression models were not used to examine associations between perceived greenness and associated outcomes.
    12. Please check for typos.
  4. The results are relatively clear, but would benefit from inclusion of additional analyses and key explanatory factors.
    1. It may be beneficial to include pairwise comparisons for sex and marital status.
    2. These variables, among others that may be in the survey, should be controlled for in the models.
    3. The authors should investigate collinearity to assess model bias.
    4. Furthermore, it’s challenging to assess the results absent adequate explanation of the methodological approach.
  5. The discussion and conclusion sections would benefit from additional engagement with the data and a grounding of the study implications for public health.
    1. In main findings (lines 212-225), the authors contend that the “difference in agreement between home and university” may concern familiarity. This completely neglects that they were measuring different constructs, and should be explored here.
    2. The authors indicate that spatial memory may be an important indicator on line 233, and thus should suggest that resident tenure would be an important measure.
    3. Finally, an overview of public health implications would be helpful.

Author Response

  1. The introduction would benefit from clarification of key terms.
    • For example, on line 35, the authors introduced the concept of “green space,” yet neglect to identify what constitutes greenspace until the methods section. This should be clarified upfront and grounded in the available literature to avoid any confusion between arguments made in the introduction and the selected research approach. Furthermore, the authors operationalized greenspace in the methods section as “bushes, meadows, forests, individual trees,” but seem to miss public parks, gardens, etc.

The relevant part was extended by examples of “green space” (line 34) and a reference was added on a meta analysis on the definition of greenness. We updated the definition of green space in the methods section and added parks and gardens to the examples (line 110).

  • In addition, the authors contend that “urban densification is responsible” for greenspace and natural feature declines (lines 34-35), yet many cities with high population densities have moved toward green rooftops, greenways, community gardens, etc. (see New York City, for example) to reintroduce greenspace, thus increasing greenspace. A more thorough review of the literature would be helpful.

We extended the part about urban densification and mentioned that best-practices already exist (line 34).

  • On line 42, the authors contend that connections between nature (are they are referring to greenspace?) and health are assumed. However, the prior paragraph cites a “growing” body of literature. In fact, there is a wealth of literature on this topic. Therefore, framing this as an assumption seems peculiar.

We softened this sentence by removing the term “assumption” (line 40).

  • The literature review does not provide sufficient evidence for key arguments.

The Perceived Neighborhood Greenness section should review a broader body of literature.

For example, the authors indicate that objective and subjective measure should be utilized when examining health benefits, yet only cite one project.

We added citations to other studies (line 50).

  • On lined 49-50, the authors state that “studies” indicate disagreement between objective and perceived greenness and walking, but only cite one paper. Please include additional evidence if referring to multiple studies.

This part was adjusted accordingly to “one study” (line 51).

  • The Neighborhood Greenness in Different Environmentssection would also benefit from a more extensive literature review.

The authors indicate that “most” research concerns home neighborhoods and unspecified areas. They seem to neglect the body of literature that examines connections to greenspace in other contexts. It would be helpful to review this in addition the two papers they cite concerning campus environments.

We extended this part of the paper and included new references for studies focusing on the workplace, schools, hospitals and retirement homes (line 66).

  • Finally, there are several awkward sentences in this section that would benefit from additional editing.

We made the section clearer by rephrasing some of the sentences.

  1. The methods section would benefit from additional clarification.
    • Please note there are several awkward sentences that make this section difficult to read.

We tried to improve the language throughout the section.

  • Please operationalize “greenness” upfront.

See the very first comment.

  • It may be helpful to include a table of all “greenness” features included in the study.

In table 1, we listed all measures related to greenness in our study. We feel that presenting data that are more specific would not be useful for the readers, since we have not looked at more specific greenness features in this paper. We do mention the specific features that are included in our overall measure of greenness in the text as follows: “The attribute table of this database covers the attribute values, built space, green space, bushes, trees, water (blue space) and other built characteristics as a summary value (e.g. sidewalks).” (line 142)

  • As the authors indicated that public transportation and bicycles were the primary means of transportation, the rationale for a 300m buffer of public transportation stops needs to be further explicated. Citing the literature would be helpful.

We extended our reasoning for choosing the buffer zone based on the distances to public transport stops and added supporting literature (line 130).

  • Other researchers may wish to replicate this study, so it would be helpful to further explain methods for delineating greenspace within 300m buffers. For example, were these shapefiles that could be examined in geospatial software?

The information on the dataset we received was extended accordingly (line 139).

  • While perceived greenness measures are published elsewhere, since they are critical to this study, it would be helpful to include here.

We added a more detailed description of the perceived greenness measures.

  • On line 138, the authors indicate that their survey concerned perceived “quality and accessibility.” However, the more pertinent language on the survey concerns “presence,” as the objective measure seems to measure presence of absence.

We did not change the wording as recommended, but we extended the section and added the new variable “perceived presence of greenness” (line 151). See also the other comments concerning the presence or absence of greenness (3.8 and 5.1).

  • A major methodological flaw is that that the authors appear to measure very different constructs with the objective (i.e., presence or absence of “greenness”) and perceived (quality, accessibility, presence of “greenness”) measures of greenness. This approach requires further clarification, as the objective measures do not account for quality and accessibility, only whether there were green features within 300m of stated locations. The authors do recognize this on lines 142-143, but the method for rectifying this (quintiles) does not address the major issue. I might suggest that they only use the presence score from the “perceived” measures. At the very least, provide some evidence (literature) for using the stated methodology.

We adjusted the section “Comparison of Objective and Perceived Greenness Scores” to make clear why we chose this approach. We did not find literature on our specific approach and context, but we made essential additions to the paper by including a new variable “perceived presence of greenness” from three items asking about the subjective presence of greenness. The original “perceived greenness” variable was kept. Therefore, we were able to make clearer why this methodology is meaningful in this context.

  • Please not that calculating BMI based on height and weight does not account for muscle mass and bone density, and thus should not be used as an indicator of overweight/obesity outside of the clinical context.

We acknowledge this weakness of BMI in the discussion. (line 346).

  • Did the authors investigate participant tenure at their current residence? This would be an important confounder.

This variable has not been assessed in the research project.

  • Please explain why linear regression models were not used to examine associations between perceived greenness and associated outcomes.

We did previously report on linear regression models, in which gender, age and income were included as possible confounding variables. We did run the analyses, and the results are essentially the same.

  • Please check for typos.

We checked the paper and adjusted a few words and phrases (e.g. m2 > m2, participant’s homes > participants’ homes).

  1. The results are relatively clear, but would benefit from inclusion of additional analyses and key explanatory factors.
    • It may be beneficial to include pairwise comparisons for sex and marital status.

We included two two-way mixed-ANOVA models controlling for these variables on perceived and objective greenness.

  • These variables, among others that may be in the survey, should be controlled for in the models.

As stated in 3.11, we already ran these analyses with confounders (regression models) without major changes in the results. Rerunning ANOVA models using the same confounders as covariates and using these confounders in the regression models as well did not change the outcomes. What is more, collinearity was not of concern (below) and no effects were obtained in the pairwise comparisons. We have added this to the text of the results section.

  • The authors should investigate collinearity to assess model bias.

Supplementary tables with collinearity statistics and a short section in the results was included.

  • Furthermore, it’s challenging to assess the results absent adequate explanation of the methodological approach.

We addressed this issue by extending the Methods-section according to the reviewer’ comments.

  1. The discussion and conclusion sections would benefit from additional engagement with the data and a grounding of the study implications for public health.

  • In main findings (lines 212-225), the authors contend that the “difference in agreement between home and university” may concern familiarity. This completely neglects that they were measuring different constructs, and should be explored here.

We discussed the implications of our approach in more detail in respect of the changes made in the analyses (line 301).

  • The authors indicate that spatial memory may be an important indicator on line 233, and thus should suggest that resident tenure would be an important measure.

A new sentence was added, stating that this variable would be important to be controlled (line 299).

  • Finally, an overview of public health implications would be helpful.

We added a new section at the end of the prospect, briefly discussing the implications of this study (line 388).

Reviewer 2 Report

The reviewer appreciates the difficulty on trying to find a relationship on perceived and actual greenness and trying to relate to mental health or other parameters.  GIS is a very useful tool and the process to acquire the actual greenness by eliminating "green spaces" not visible by most people is good.  I do not know how the questionnaire defined the green spaces to those responding- for example, it may be that some students surround themselves with plants indoors and grow small gardens close to their dwelling spaces increasing the perception of well-being.  Also, were there any questions about how "optimistic" the person answering the survey is?  There are individuals that will have a good, positive outlook on life regardless of where they live.

Overall- the approach and the paper makes sense.  Looking forward to future analyses. 

Additional Comments:
Strength of the manuscript- its scientific approach to define objective greenness using a GIS tool to identify green spaces in the traditional way but taking the extra step to eliminate those areas that were not visible to most people. Weakness- not sure if it is a weakness as the specific questionnaire does not provide the question but I would say that it may be missing the fact that some students have their own "indoor" green spaces by growing plants and may also have small gardens increasing the perception of well-being. It also does not account for the fact that some people are inherently "pessimistic" and others are "optimistic" influencing their overall vision of the world around it. The authors may want to provide some additional information about how they can account for the "indoor" green environments and the personality traits of pessimistic/optimistic. In addition, the authors stated that the students may spend more time in their dwellings than at campus. This may not necessarily be true for a full academic schedule. The difference in "perception" may be due to the fact that the campus may be like "work" and home is where they decompress and built their "sanctuary" in other words- home is where the participants may consciously build an environment conducive to better well-being. Was this considered? Overall the topic is not easy as trying to make some correlation between the subjective and objective view of any parameter is difficult. I participated in a study that had questionnaire questions trying to figure out if "green spaces" made people happy. It did not account for people that hated the outdoors due to allergies or people that grew up in the desert- the excessive greenness of other environments was suffocating. This paper is well structure and the approach is good.

Author Response

Strength of the manuscript- its scientific approach to define objective greenness using a GIS tool to identify green spaces in the traditional way but taking the extra step to eliminate those areas that were not visible to most people.

Weakness- not sure if it is a weakness as the specific questionnaire does not provide the question but I would say that it may be missing the fact that some students have their own "indoor" green spaces by growing plants and may also have small gardens increasing the perception of well-being. It also does not account for the fact that some people are inherently "pessimistic" and others are "optimistic" influencing their overall vision of the world around it. The authors may want to provide some additional information about how they can account for the "indoor" green environments and the personality traits of pessimistic/optimistic.

We added an additional section at the end of “Strengths and Limitations” to account for this the important confounder “indoor greenness” (line 362). We also shortly discussed natural “pessimists” and “optimists” (line 335).

In addition, the authors stated that the students may spend more time in their dwellings than at campus. This may not necessarily be true for a full academic schedule. The difference in "perception" may be due to the fact that the campus may be like "work" and home is where they decompress and built their "sanctuary" in other words- home is where the participants may consciously build an environment conducive to better well-being. Was this considered?

We briefly discussed “familiarity” with home environments influencing the results, which is similar to this point. However, the general university-system in Austria, especially at the University of Graz, is quite “open”, leaving a lot of free-time for learning and working at home, while time of presence in class is kept very low. In Austria, there is a distinction between “Fachhochschulen”, which have equally-ranked degree-systems as universities, and universities. “Fachhochschulen” are based on the classical teaching in presence and fixed full-time schedules, whereas university schedules are not fixed and depend more on learning at home. Almost all participants in the study can be traced back to the University of Graz or one of the other two major universities in Graz, which is why we made this assumption.

Overall the topic is not easy as trying to make some correlation between the subjective and objective view of any parameter is difficult. I participated in a study that had questionnaire questions trying to figure out if "green spaces" made people happy. It did not account for people that hated the outdoors due to allergies or people that grew up in the desert- the excessive greenness of other environments was suffocating. This paper is well structure and the approach is good. 

We acknowledge that there are certainly aspects that we did not cover in our study, such as allergies, and thank the reviewer for the positive comments.

Round 2

Reviewer 1 Report

Thank you for making the suggested changes.